# Uncovering Latent Memories in Large Language Models

**Sunny Duan**
Brain and Cognitive Sciences
MIT
sunnyd@mit.edu

**Mikail Khona**
Physics
MIT
mikail@mit.edu

**Abhiram Iyer**
EECS
MIT
abiyer@mit.edu

**Rylan Schaeffer**
Computer Science
Stanford University
rschaef@cs.stanford.edu

**Ila Rani Fiete**
Brain and Cognitive Sciences
MIT
fiete@mit.edu

## Abstract

Frontier AI systems are making transformative impacts across society, but such benefits are not without costs: models trained on web-scale datasets containing personal and private data raise profound concerns about data privacy and misuse. Language models are trained on extensive corpora including potentially sensitive or proprietary information, and the risk of data leakage, where the model response reveals pieces of such information, remains inadequately understood. Prior work has demonstrated that sequence complexity and the number of repetitions are the primary drivers of memorization. In this work, we examine the most vulnerable class of data: highly complex sequences that are presented only once during training. These sequences, often containing the most sensitive information, pose a considerable risk if memorized. By analyzing the progression of memorization for these sequences throughout training, we uncover a striking observation: many memorized sequences persist in the model's memory, exhibiting resistance to catastrophic forgetting even after just one encounter. Surprisingly, these sequences may not appear memorized immediately after their first exposure but can later be "uncovered" during training, *even in the absence of subsequent exposures* – a phenomenon we call "latent memorization." Latent memorization presents a serious challenge for data privacy, as sequences that seem hidden at the final checkpoint of a model may still be easily recoverable. We demonstrate how these hidden sequences can be revealed through random weight perturbations, and we introduce a diagnostic test based on cross-entropy loss to accurately identify latent memorized sequences.

## 1 Introduction

Frontier AI models are trained on vast web-scale datasets (Touvron et al., 2023; Gemini Team et al., 2023; OpenAI et al., 2023; Brown et al., 2020). The sizes of these pretraining corpora enable fluency, knowledge about various domains (AlKhamissi et al., 2022; Guu et al., 2020), and the ability to perform in-context learning (Brown et al., 2020). However, these datasets often include proprietary, copyrighted, or otherwise private information (Smith et al., 2023; Karamolegkou et al., 2023; Bordt et al., 2024; Duan et al., 2024; Staab et al., 2023; Shi et al., 2023; Tang et al., 2023; Zanella-Béguelin et al., 2019), which is problematic because LLMs have been shown to possess a vast capacity for detailed memorization. Specifically, with appropriate prompting, LLMs can regurgitate verbatim text from their training corpora.

Prior work has found that even sequences encountered early in training can be extracted from the model, long after they have been encountered (Biderman et al., 2023b). One possible cause of this is that memorized sequences appear multiple times within the corpus, allowing the network to reinforce and store this data in its weights. Our findings confirm that repeated sequences constitute the majority

of the memorized content. However, we also find many sequences which are encountered only once during training but are memorized by the model and persist in the model's memory throughout the training process. Many of these memories may seem forgotten during certain stages of training but are later recalled without additional exposure, indicating they remain encoded in the model's weights. These 'latent' memories present significant challenges, as they are not easily detected by current memorization metrics, raising the question of how to effectively identify and quantify memorized training data in large language models.

## 1.1 CONTRIBUTIONS

This work provides significant insights into the dynamics and mechanics of memorization in language models during pretraining, contributing to the broader understanding of data privacy and security within machine learning. Our primary contributions are as follows:

- **Quantification of Memorization Susceptibility**: We systematically evaluate how the statistical characteristics of training data, specifically sequence complexity and repetition, influence the likelihood of memorization in language models. Our findings demonstrate that the probability of memorizing a sequence scales logarithmically with its repetition in the training data as well as the complexity of the sequence under consideration. These results extends prior work characterizing which sequences become memorized (Prashanth et al., 2024; Tirumala et al., 2022).

- **Stationarity of Memorized Sequences**: By analyzing how memorization changes throughout training, we discover that the memorization status of sequences remains largely stationary after initial exposure, despite not being re-encountered. We find that for many sequences, memorized sequences may disappear and re-appear in the model's output without repeated exposure. This indicates a fundamentally persistent property of the memory, revealing how the state of memorized sequences is preserved and how subsequent training only modifies the model output.

- **Latent Memorization and Recovery**: We identify the presence of "latent" memorized sequences, which are not evident at certain checkpoints but can be uncovered later in training or through controlled perturbations. Our experimental results show that adding random Gaussian noise to model parameters can recover these latent memorized sequences, supporting the hypothesis that further training acts as random additive noise rather than fundamentally altering the memorization state.

- **Development of a Diagnostic Test**: We propose a novel diagnostic test for uncovering latent memorized sequences by analyzing their cross-entropy loss. This test provides a practical tool for detecting and mitigating potential data leakage in deployed language models.

Our study underscores the risks associated with data leakage in language models, emphasizing the need for more robust mechanisms to ensure data privacy. The persistence of memorized sequences poses a challenge for the prevention of data leakage. By characterizing the nature of memorization as well as the nature of these latent memorized sequences, we elucidate possible mechanisms of how sequences become memorized and offer practical solutions for mitigating data privacy risks, and developing safer and more trustworthy models.

## 2 METHODOLOGY

### 2.1 PROPERTIES OF PRETRAINING DATA RELEVANT TO MEMORIZATION: REPEATS AND COMPLEXITY

Previous studies have identified that the number of repeats of a sequence affects whether it will be memorized, with more frequently occurring strings being more likely to be memorized (Carlini et al., 2020; Razeghi et al., 2022; Biderman et al., 2023a). Consequently, a starting property to measure is the **number of repeats** of specific strings in the pretraining corpus.

In our work, we also consider a second and newer property: the **complexity** of specific strings. Our decision to do so is motivated by previous studies (Carlini et al., 2020) which identified a prevalent class of easily memorized data: simple sequences composed of repeated patterns, numbers, or other

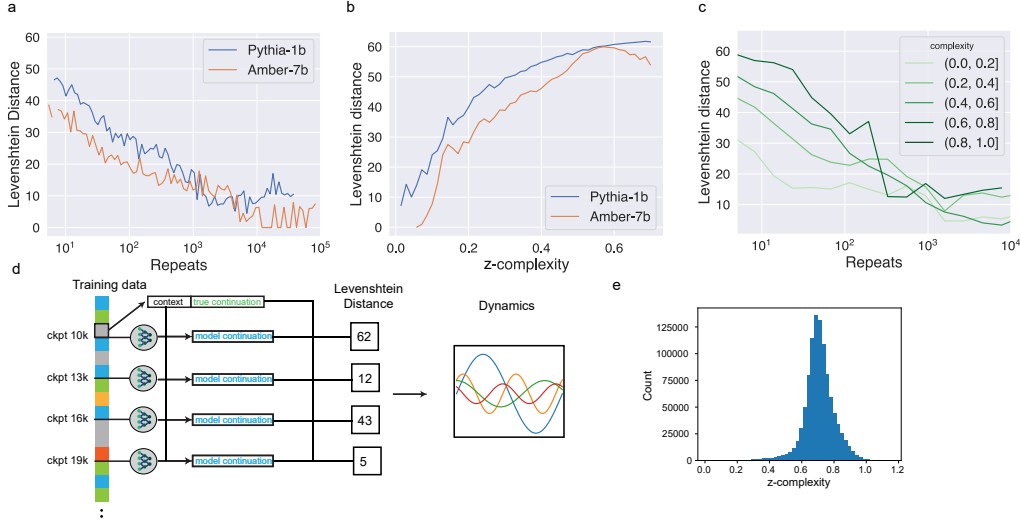

Figure 1: **Data statistics and the probability of memorization a.** Plot of average kl-LD as a function of the number of times the sequence is repeated in the dataset for Pythia-1b and Amber-7b **b.** Average kl-LD as a function of the Z-complexity of the sequence. **c.** Relationship between kl-LD and repeats for different complexity levels. **c.** Schematic of pipeline for analyzing memorization over time. All samples were selected from early on in training. Various model checkpoints are selected and evaluated to determine if early training sequences are still memorized. The changes in k-LD over time are used in our analysis to evaluate how the memorization of these sequences changes throughout training. **d** Distribution of z-complexity over all of the data.

straightforward patterns. While models readily learn these samples, they often lack substantive content and are unlikely to represent sensitive information. Thus, it is important to distinguish between memorization of these trivial sequences from more complex and informative sequences.

In order to quantitatively measure the complexity of specific strings, we use modern compression algorithms to determine the extent to which sequences have a smaller description than the original sequence. To calculate the complexity of a sequence we define a metric, z-complexity, which is the ratio between the compressed sequence length to the original sequence length. This metric contains values from $0$ to $1$ and is efficiently computable using the `zlib` package in Python.

## 2.2 QUANTIFYING MEMORIZATION

Several different definitions have been put forward to quantify memorization in language models. Intuitively, a memorized sequence is a training sequence which can be reproduced given the right context (Carlini et al., 2022; Schwarzschild et al., 2024). One popular definition of memorization is $kl$-**memorization** (Carlini et al., 2022). $kl$-memorization is evaluated by considering a sequence of length $k + l$. The first $k$ tokens are presented to the model as context. The model is used to generate a continuation of length $l$ via greedy (i.e., temperature $= 0$ decoding). The model's continuation is compared to the "true" continuation, and a sequence is said to be $kl$ memorized if the model's output exactly matches the true continuation.

While undoubtedly useful, we introduce and utilize a new metric for measuring memorization. $kl$-memorization is an overly strict such that even a single-token deviation from the true continuation may cause us to misclassify a sequence as forgotten; in many cases, the model may make small errors such as inserting or modifying a single token. We identified and provide several examples in Table 1. In order to to be more robust to small changes in the learned sequence, we propose a modification of $kl$-memorization by introducing **kl-Levenshtein distance (kl-LD)**.

**Definition 2.1** (kl-LD distance). Let $S = (s_1, s_2, \ldots, s_n)$ be a sequence of tokens. We denote the first $k$ tokens as the context $C = (s_1, \ldots, s_k)$ and the last $l$ tokens as the target $T = (s_{k+1}, \ldots, s_{k+l})$. The

model is provided these context tokens and produces a predicted continuation of $\hat{T} = (m_1, \ldots, m_l)$. We define the kl-LD distance as the Levenshtein distance (Levenshtein, 1965) between sequences $T$ and $\hat{T}$ where the Levenshtein distance is the minimum number of (token) insertions, deletions or substitutions that must be performed on $T$ to obtain $\hat{T}$.

We find that this is a natural measure of memorization which also provides a range of values to provide more granular insight into the strength of the model's memory, akin to how continuous metrics have been used to improve evaluations of language models (Schaeffer et al., 2023; 2024). Throughout this study, we set $k = 32$ and compare the continuation of the model with the original sequence by computing the Levenshtein distance between the next $64$ tokens.

Table 1: Model continuations at various stages in training for a few selected sequences which were complex and encountered only once during training. Minimum edits are highlighted such that character edits are highlighted in orange, deletions are highlighted in red and new characters are highlighted in green.

| Context | True Continuation | Checkpoint 10000 | Checkpoint 15000 | Checkpoint 19000 |
|---|---|---|---|---|
| 992> por favor ayudenme para instalar DBDesigner <him> hay <BluesKaj>!es \| Guest17992 | <ubottu> Guest17992: En la mayorÃxada de canales Ubuntu se comunica en inglÃ©s. Para ayuda en EspaÃ±ol, por favor entre en los canales #ubuntu-es o #kubuntu-es. <BluesKaj>!es \| | <ubottu> Guest17992: En la mayorÃ a de canales de Ubuntu se comunica sÃ²lo en inglÃ©s. Para busca ayuda en EspaÃ±ol, por favor entrera en los canales #ubuntu-es o #kubuntu-es. <BluesKaj>! es l | <ubottu> Guest17992: En la mayorÃ a de canales de Ubuntu se habla sÃ³lo en inglÃ©s. Si busca ayuda en espaÃ±ol o charlar entra en el canal #ubuntu-es. Escribe "/join #ubuntu-es" | <ubottu> Guest17992: En la mayorÃ a de los canales de Ubuntu, se habla sÃ³lo en inglÃ©s. Si busca ayuda en espaÃ±ol entre al lar entra en el canal #ubuntu-es, escriba "/join #ubuntu-es",( |
| , findings, beliefs, or experiences on those topics or products. The views and opinions expressed on CateTheOkay.com are purely my own. Any product | claim, statistic, quote or other representation about a product or service should be verified with the manufacturer, provider or party in question. CateTheOkay.com doesn't contain any content which might present a conflict of interest. | claim, statistic, quote or other representation about a product or service should be verified with the manufacturer or provider. Comments. I have a question. I have a friend who is a teacher and she is a teacher. She is a teacher and she is a student. She is a student and she is a | claim, statistic, quote or other representation about a product or service should be verified with the manufacturer or provider or party in question. CateTheOkay.com is not affiliated with, endorsed by, or sponsored by the Coca-Cola Company. CateTheOkay.com is not affiliated with, endorsed by, | claim, statistic, quote or other representation about a product or service should be verified with the manufacturer or provider or party in question. I am not a doctor, pharmacist, or registered dietitian. I am not a registered dietitian. I am not a registered dietitian. I am |

### 2.2.1 ANALYZING REPEATED SAMPLES

In this study, we seek to understand both how repeated encounters of a sequence during training drives memorization and also how sequences which are encountered only once are retained by the model. To this end, we analyze where training sequences are repeated throughout the course of training. In our study, we focus on the $l$ portion of the sequence. For this study, we fixed $l$ to 64 tokens. Given a target sequence, we compare the target sequence with all of the training sequences which were presented to the model during the period of training under consideration. We compute the largest subsequence match between the target and every individual training example and call a training example a "repeat" if there was a sub-sequence match of length $30$ or longer.

### 2.3 LANGUAGE MODELS

In this study, we largely focused on the Pythia 1B language model (Biderman et al., 2023a), which was trained on 300B tokens from the Pile (Gao et al., 2020). For selected experiments, to ensure our results hold on other language models, we reproduced our results using a second model, Amber-7B (Liu et al., 2023). We selected these two models as they were large, high performing models complete with fully reproducible data sequences and frequent checkpoints. As in previous works (Biderman et al., 2023a), all experiments were run with the models run with half precision (`float16`) and temperature 0.

### 2.4 DATASETS

In this work, we use the deduplicated versions of the Pile (Gao et al., 2020) as well as the Amber dataset which is a combination of the RedPajama V1, RefinedWeb and StarCoderData datasets (Computer, 2023; Li et al., 2023; Penedo et al., 2023), all of which employ deduplication. In the early part of our work we also employ the standard Pile which did not use de-duplication in order to

observe the effects of repeated exposure on memorization. In the latter part of our work we focus on the deduplicated versions of the datasets in order to eliminate the influence of repeated exposure on our analysis of memorization.

### 2.4.1 LANGUAGE MODEL CHECKPOINTS

In our analysis, we used checkpoints from every 3k training steps between from step 10k-43k in Pythia-1B and every 10 checkpoints of Amber-7B, corresponding to roughly 1.7 million training examples between revision 100 to 350. These selections were checkpoints from each model which represented a sizable portion of training. These were chosen to be offset from the beginning of training to avoid artifacts or initial transients from random initialization, learning rate warmup and other peculiarities from initial phases of training.

## 3 EXPERIMENTAL RESULTS

### 3.1 DATA STATISTICS PREDICT MEMORIZATION

We analyze two primary drivers of memorization during training: sequence complexity, and the number of repetitions. Previous work showed that the probability a training string can be extracted from a model is related to the model size and number of repetitions (Carlini et al., 2020); we find that this relationship is true in the models we analyzed as well (Figure 1a). Additionally, we found that the complexity of the string itself was a strong predictor of whether it would be memorized (Figure 1b): strings with smaller z-complexity had smaller kl-Levenshtein distance (kl-LD), meaning simpler strings are more easily memorized. Interestingly, recent work showed that pretraining language models on data with lower z-complexity causes the training losses to decrease more rapidly (Pandey, 2024); our results here suggest an explanatory mechanism: with more compressible data, the model can memorize the data more quickly. Furthermore, we found that for strings of different complexity exhibited different memorization curves (Figure 1c), whereby lower complexity strings were more easily memorized with fewer repeats. Both of these factors influenced the memorization probability with a log-linear relationship. Since highly sensitive information is likely contained in complex and rare training sequences, we focus our efforts on these sequences. **In the rest of this work, we restrict our analys to sequences which are presented once and have high complexity** ($> 0.8$).

### 3.2 DYNAMICS OF MEMORIZATION

In order to produce a more complete picture of how successive training affects the state of memorized sequences within our model, we analyze how the kl-LD changes throughout the course of training for individual sequences. We select sequences early on in training and evaluate how the memorization status of these sequences evolves throughout training (Figure 1a). In this section we utilize the deduplicated version of the Pile dataset (Gao et al., 2020) as well as the Amber LLM360 dataset which also uses deduplication in order to remove the effects of repeated exposures. We filter these sequences so that only sequences which have a z-complexity of $0.8$ or higher are included in our analysis. Additionally, we employed our own duplication detection scheme which eliminated sequences which had a sub-sequence match of length 30 or longer and rerun our analysis using these sequences (Figure S13).

Surprisingly, we find that the memorization status of a sequence is largely stationary throughout training. After the initial checkpoint, the kl-LD of the sequences fluctuate (Figure 3) but do so in a way which is stationary across training (Figure 2a). This is consistent across both Pythia-1b and Amber-7b models. This is reflected in the individual trajectories, and also in the overall mean of the population which shows no clear trend as training progresses. Furthermore, unlike a random walk, we see that the variance of the does not grow over time, but remains fixed. We can quantify this by running a variance ratio test A.2, where the variance of a random walk is expected to grow linearly. We can reject the null hypothesis that our data is generated from a random walk with $p < 10^{-8}$ for the samples drawn from Pythia-1b and Amber-7b. This is indicative of a mean reversion tendency of the dynamics and demonstrate the stability of the memories within the model weights. Additionally, we observe that the changes in the kl-LD between consecutive checkpoints are symmetric (Figure 2c) and roughly follow a laplace distribution (Figure 3). This again confirms the counter-intuitive property of sequences to become memorized as often as they are forgotten. Notably, the model is able

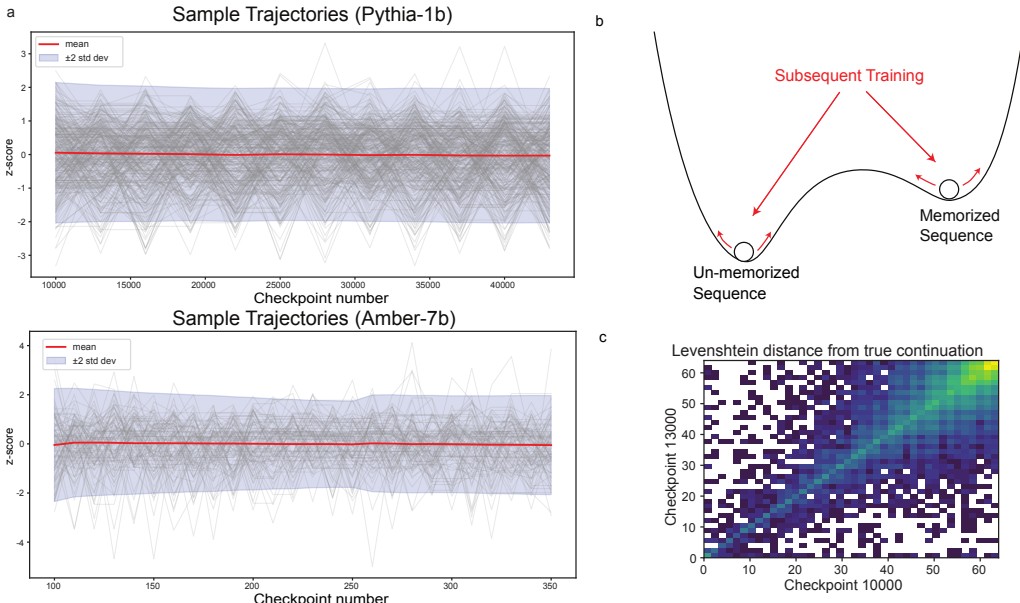

Figure 2: a. Visualization of individual samples and the change in the memorized length during training. Grey lines are subsampled single sequence trajectories throughout training. Each sequence was normalized such that the distribution of memorization lengths was mean 0 and variance 1. The red line denotes the mean and shaded area denotes region of two standard deviations of the kl-LD of all sequences at a single point in time. Notably, the distribution at each timestep is the same for all checkpoints. This is in contrast to both the expected exponential decay behavior exhibited by models which experience catastrophic forgetting as well as the linear growth of variance which is expected of processes exhibiting random walk behavior. b. Conceptual schematic of how memorized sequences may be stabilized during training in order to resist the interference from weight changes caused by subsequent training. c. Joint distribution of kl-LD during checkpoints 10k and 13k. Color is the log of the number of sequences in each bin. The vast majority of sequences are not memorized in either checkpoint.

to recall memories which, at one point in time, appeared to be forgotten, despite never encountering that sequence again.

The stationarity of the memorization status of these sequences indicates that the memorized sequence is stable throughout time, but this is in conflict with the fact that the model weights are constantly evolving. This stability in the presence of noise is indicative of a stabilizing mechanism by which the encoding of the sequence memory is preserved by some restorative process illustrated in Figure 2b where the memorized sequence becomes a stable fixed point in the weight space of the model under training dynamics. Since this is not true of all sequences, but only the few which exhibit this persistent memorization, it may point to a phase transition that occurs when the sequence is first encountered.

## 3.3 LATENT MEMORIZATION AND RECOVERY OF LATENT MEMORIES

Since some sequences exhibited seemingly random variations in their memorization state across different checkpoints, we hypothesize that these sequences remain memorized but are not be visible at a given checkpoint and are "latent" memorized. Indeed, we found many sequences which were not memorized at the initial checkpoint (10k) but exhibited memorization by checkpoint 19k (Table 1).

For these sequences, the nature of the random changes shown in Figure 2a indicate the form of a random walk. We hypothesize that the process of training in frontier AI models acts as random noise on the weights with respect to the memory of the sequence. Thus, simply perturbing the weights with random noise should produce similar effects as training.

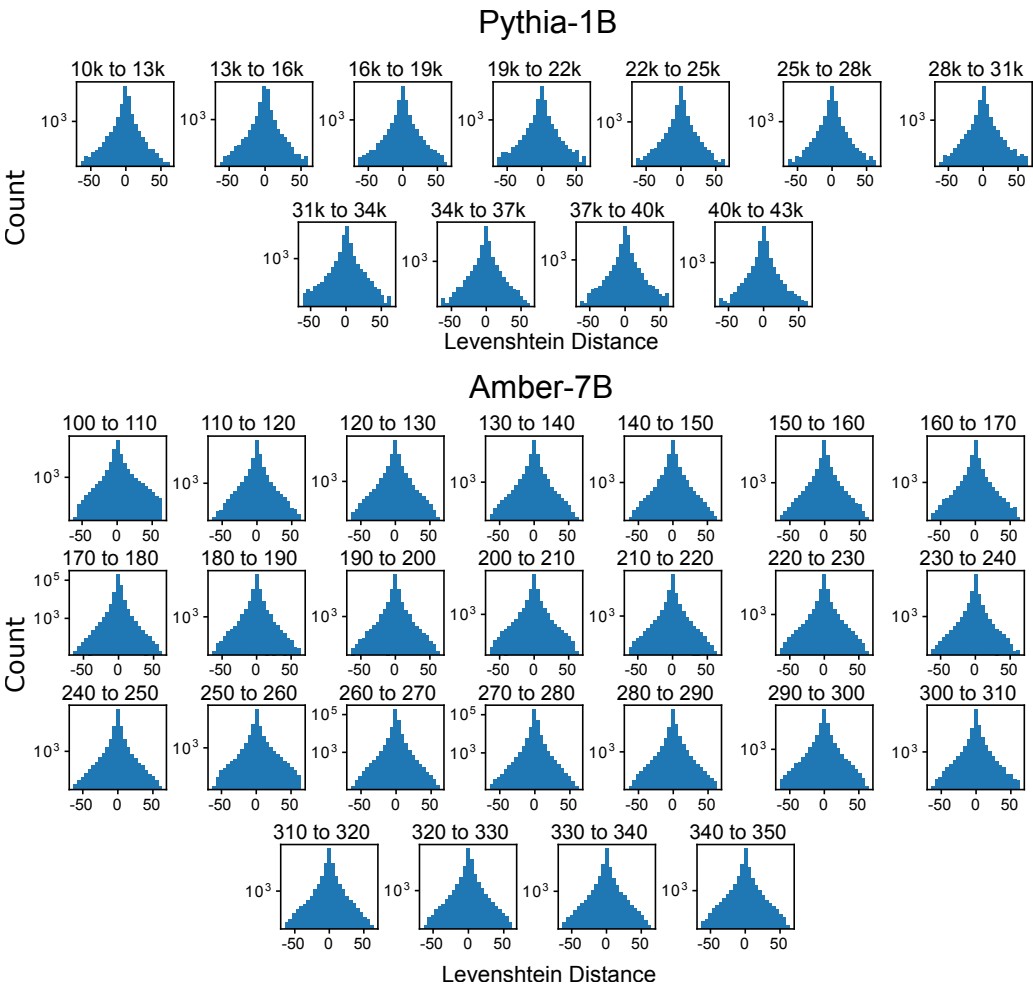

Figure 3: Visualization of how memorization status, measured by kl-LD, changes between consecutive checkpoints for both Pythia-1B and Amber-7B.

We find that this prediction is true. We randomly perturb the model weights by adding a small amount of random gaussian noise (of magnitude $2 \times 10^{-3}$) to each of the weight parameters. We repeat this process 200 times and find the perturbation which yields the lowest kl-LD (Figure 4a). Notably, in the high dimensional weight space, it is difficult to reproduce arbitrary sequences using random weight perturbations, thus the recovery of memorized sequences must be due to intrinsic factors of how the memory is encoded in the weights.

We find that sequences which were "latent" memorized are able to be recovered using random perturbation (Figure 4bc). In contrast, sequences which were not memorized during the period of consideration could not be recovered. As a control, we also selected sequences which were not presented to the model yet, and observed that their distributions closely matched those which were encountered by not memorized by the model (Figure 4b). Furthermore, we found that the perturbations yielded memorization patterns which closely matched that of the model at a later point in training. These observations support the view that with respect to a memorized sequence, subsequent training acts similar to random noise perturbations to the model weights.

As an additional control, we attempted to recover the latent memorized sequences by sampling from the model at different temperatures. Since these sequences are stored in the model weights, it may be the case that the model would simply reproduce the target sequences if prompted enough times. We tried four different temperatures and sampled 200 different sequence continuations from each

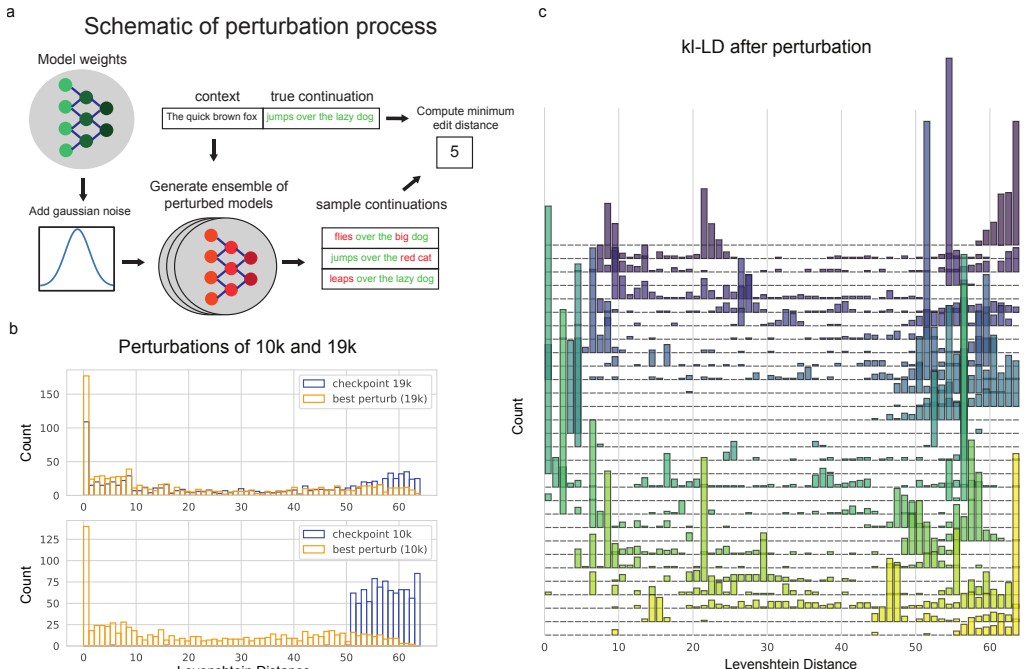

Figure 4: **a**. Schematic of how model weights are perturbed in order to extract memories from the model. The same context is given to 200 perturbed modesl and the best continuation is chosen. **b**. Comparison of the distribution of best achievable kl-LD by perturbing the model weights. Data points were selected such that they were un-memorized (kl-LD > 50) at 10k but we're memorized (kl-LD < 10) at some point during the next 10k training steps. Top panel is the histogram of the perturbations of the checkpoint at 19k and bottom is 10k. Notably, the perturbations cause the 10k model distances to match the distribution of the 19k model, and perturbing the 19k model does not have a significant effect. This is indicative of how model training mimics random noise with respect to the memorization status of the sequences. **c.** Visualization of the Levenshtein distances from the target for various weight perturbations. Each row is a single sequence, and the heights of the bars correspond to the number of perturbations which resulted in a Levenshtein distance of that corresponding bin.

of the temperatures. We find that the this method fails to recover the latent memories that weight perturbation was able to produce (Figure 5b).

"Latent" memorized sequences pose a significant risk for leakage since they are not easily detectable from evaluating kl-memorization of those sequences. To this end, we discovered that these "latent" memorized sequences had significantly lower cross entropy loss when evaluated by the model (Figure 5c), thus simply evaluating the likelihood of those sequences using the trained model is a natural diagnostic for detecting these "latent" memorized sequences.

## 3.4 RELATED WORK

Extracting memorized sequences from language models is an area of high interest. Early work established that it was possible to extract sensitive data including phone numbers, URLs and personal information from trained language models (Carlini et al., 2020). Other studies injected canaries to determine what aspects of the training process contributed to whether a sequence is extractable (Henderson et al., 2017)(Thakkar et al., 2020). More recent work have extended this to investigate how these properties scale with model size and data statistics (Carlini et al., 2022). This has motivated the use of deduplication, which in addition to reducing the chance of data leakage (Kandpal et al., 2022), also has been shown to improve sample efficiency and improve evaluation (Lee et al., 2021).

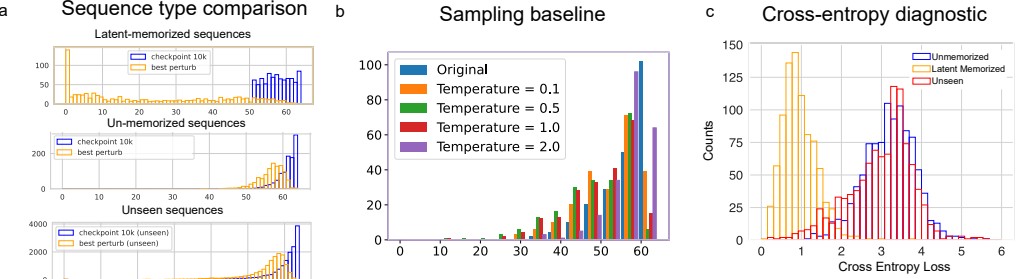

Figure 5: **a.** Comparison of using perturbations to evoke a target sequence for three different classes of sequences. In the top panel, we examine the sequences which are "latent" memorized. In the middle panel, we find sequences which weren't memorized during training and in the bottom panel, we analyze sequences which were encountered later in training but were not encountered by the model. We not that perturbing the weights is only able to evoke sequences which are "latent" memorized. **b.** Attempts at evoking latent memories using 200 samples at various temperatures. None of the temperatures were able to reliably recover the latent memories. **c.** Comparison of the cross entropy losses of sequences separated into the three different classes of sequences analyzed in b. The cross entropy losses of "latent" memorized sequences are much lower.

The definition of memorization is also still debated and various approaches to quantifying memorization have been made (Zhang et al., 2021; Feldman & Zhang, 2020). A variety of attacks have been designed to extract memorized sequences using designed prompts (Thakkar et al., 2020) and model activation perturbations (Kassem et al., 2024).

More generally, the notion of membership inference has been studied as a way to determine whether a given training example was part of the corpus (Shokri et al., 2016; Mireshghallah et al., 2022; Hisamoto et al., 2019), and these approaches have been applied to language models as well (Duan et al., 2024).

Forgetting has also been studied extensively in neural networks, typically in the context of preventing forgetting. (Kirkpatrick et al., 2017; Zenke et al., 2017; Chen et al., 2020). Studies have also shown that forgetting decreases with model size (Tirumala et al., 2022; Mirzadeh et al., 2021). This work has also been examined in the context of understanding what aspects about a model and the data contribute to forgetting (Toneva et al., 2018)

Finally, there has also been work studying how the training process affects the status of memorization (Tirumala et al., 2022; Prashanth et al., 2024). This work focuses on how parameters of training and size of the model affect the dynamics of training. They find that scaling the model generally leads to less forgetting. In our work, we focus on sequences which counter-intuitively do not obey the forgetting laws presented in this work and expanding on the implications of these persistent "episodic" memories.

## 4 CONCLUSION AND LIMITATIONS

We study how memorization changes throughout training and focused on sequences which occurred only once throughout training. Under these conditions, we find that rather than forgetting these sequences, the model retains them for the duration of training. This stationarity indicates a stability of the memorized sequence in weight space since the training process necessarily modifies the weights which encode the memorized sequences. We test this mechanistic view of how the training process interacts with the memorized sequence by using random weight perturbations to the model weights. These perturbations confirm that sequences which appeared to be forgotten at one point during training, may still be memorized by the model and are able to be uncovered with a small amount of random noise. We concluded by demonstrating a simple diagnostic to distinguish between "latent" memorized sequences and un-memorized sequences.

This study highlights one surprising behavior of frontier AI models and begins to uncover what mechanisms are present in the memorization behavior of these models. Our work suggest a possible mechanism of how memorized strings are sustained throughout training and further experiments are needed to confirm the underlying mechanism. Notably, further testing is required across other frontier AI models which were not considered here. Finally, we propose a mechanistic explanation for this phenomenon which requires further study to explain the cause of these persistent memories.

## 5 REPRODUCIBILITY STATEMENT

All code used for this project is available at `https://github.com/sunnyddelight/latent_memorization`.

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

## A  APPENDIX / SUPPLEMENTAL MATERIAL

### A.1  COMPUTE DETAILS

All experiments were run on a cluster with access to 16 concurrent a100 GPUs. All of the language models were run using a single GPU and multiple GPUs were used to parallelize the experiments in order to speed up progress. Searching for repeats within the dataset was performed using the library dask, using 64 CPUs distributed in a cluster, each with 32Gb of RAM.

### A.2  VARIANCE RATIO TEST

Random walks have a hallmark property of linearly increasing variance over time. We can demonstrate statistically that the sequence of memorization lengths does not follow a random walk by conducting a variance ratio test. Given a sample $\{\{X_{ij}\}_{1 \leq i \leq m}\}_{1 \leq j \leq n}$ of $m$ sequences of length $n$, we can calculate the F-statistic by taking the ratio of the variances

$$\frac{\frac{1}{m}\sum_{i=1}^{m}(X_{in} - \bar{X}_{in})^2}{n\frac{1}{m}\sum_{i=1}^{m}(X_{i1} - \bar{X}_{i1})^2}$$

which, for a random walk has an F-distribution with m and m degrees of freedom.

### A.3  LICENSES

This project used code from the Pythia project Biderman et al. (2023a) released by EleutherAI under the Apache license version 2.0. We also used the Pile dataset Gao et al. (2020) which is released under the MIT license. The Amber model was produced by LLM360, and the code and dataset are both released under Apache 2.0.

### A.4  ADDITIONAL FIGURES

We include figures which were omitted from the main paper. These provide additional details that were not central to the claims made in the paper.

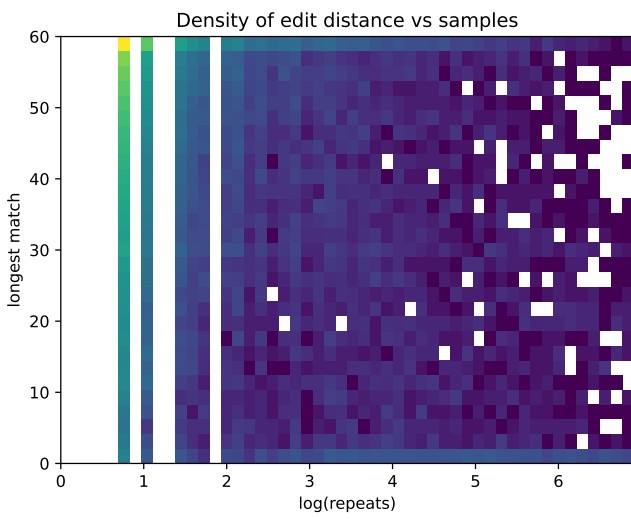

Figure 6: **Histogram of the repeats vs the edit distance** Hue is log density.

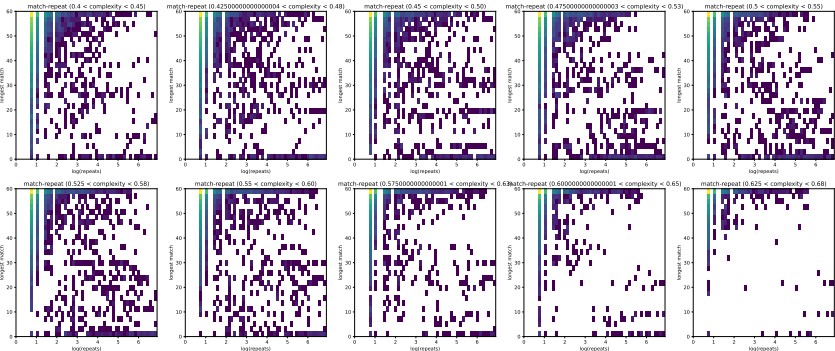

Figure 7: **Histogram of the repeats vs the edit distance split by complexity** Hue is log density.

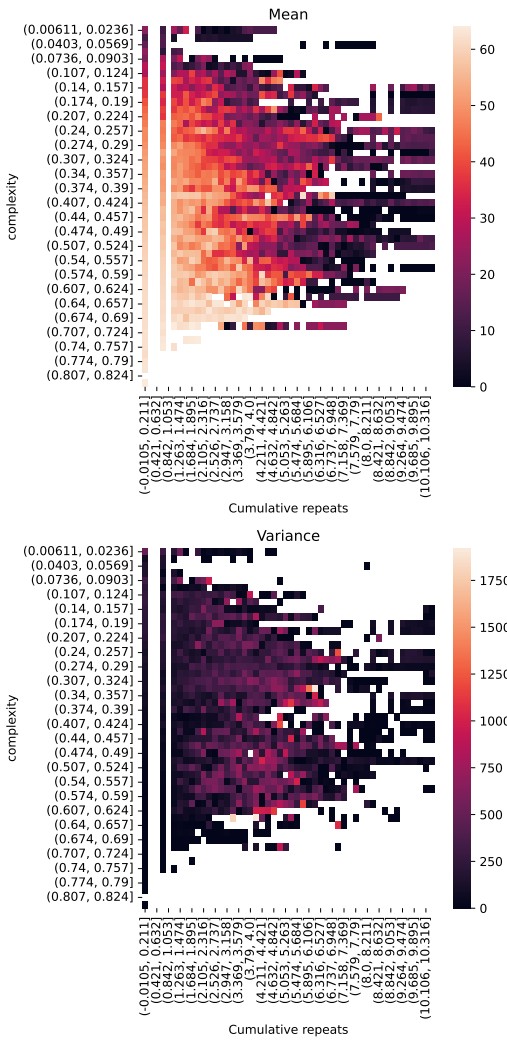

Figure 8: **Average of the kl-LD metric** kl-LD values are binned by number of repeats and complexity and the mean and variance of the samples in those bins are computed and colored.

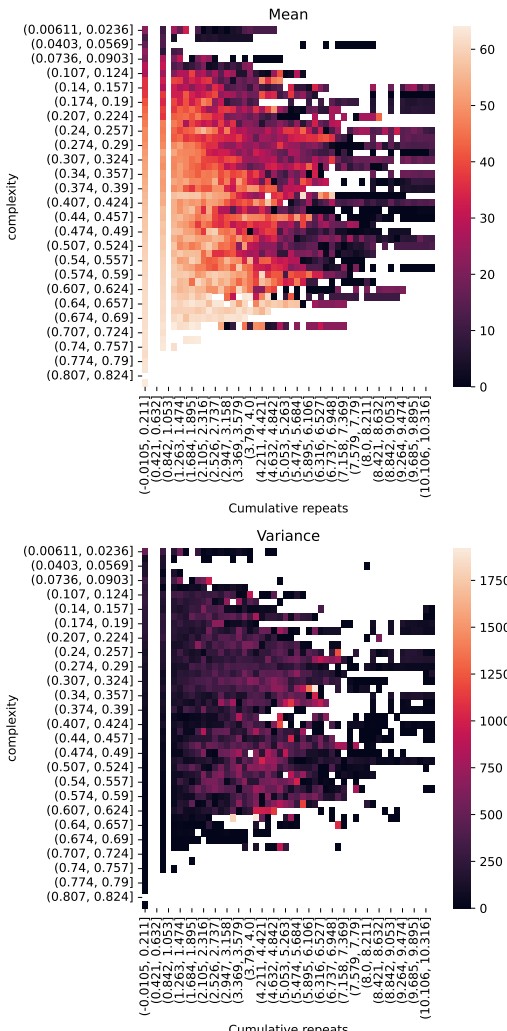

Figure 9: **Average of the kl-LD metric** kl-LD values are binned by number of repeats and complexity and the mean and variance of the samples in those bins are computed and colored.

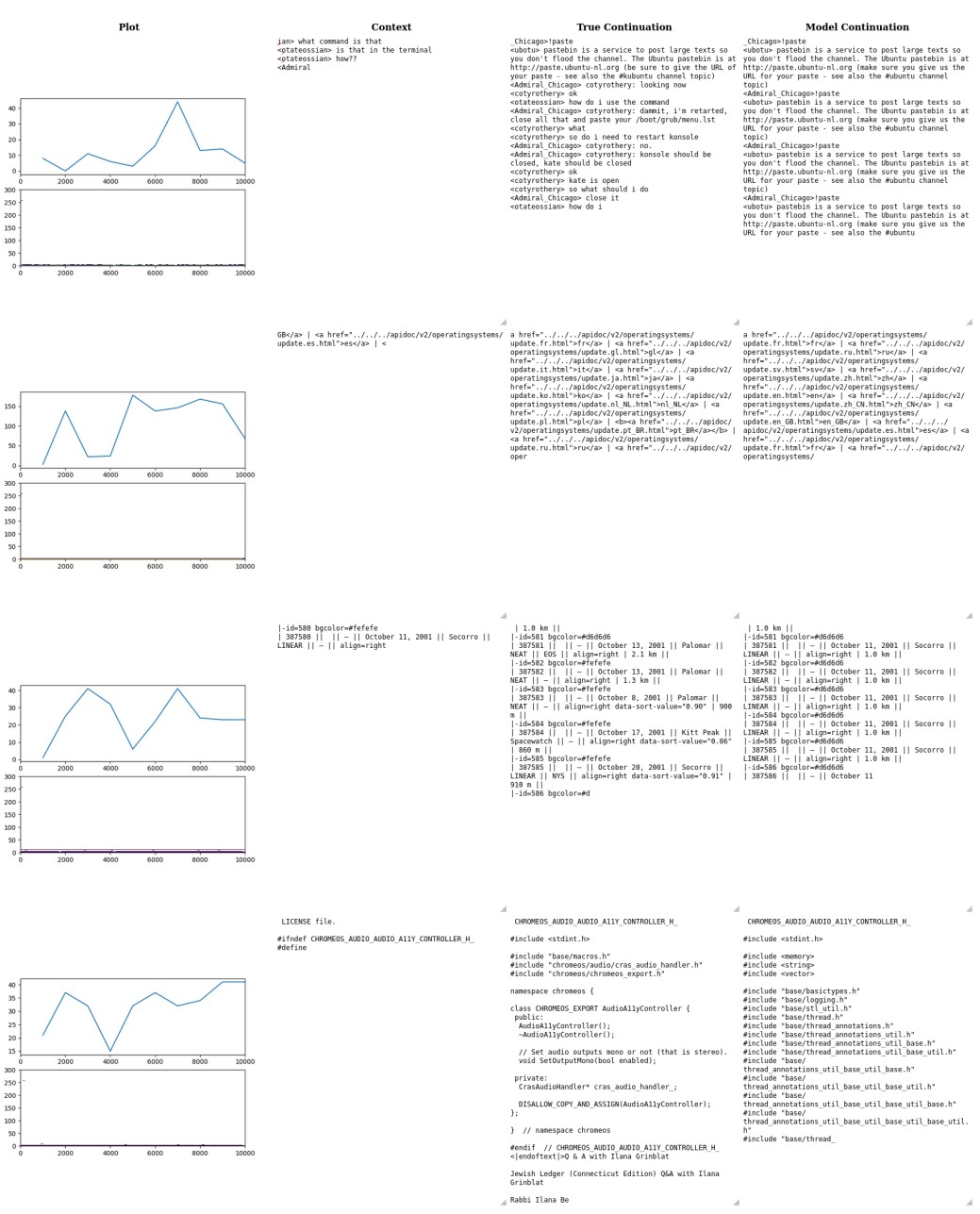

Figure 10: Examples of strings which were seen once during training. Top left plot shows the kl-LD over for different trajectories and bottom left plot is a histogram of when the examples were repeated and at what length with the time on the x axis and the length of the repeat on the y axis. The text of the context, true continuation and model continuation are shown as well.

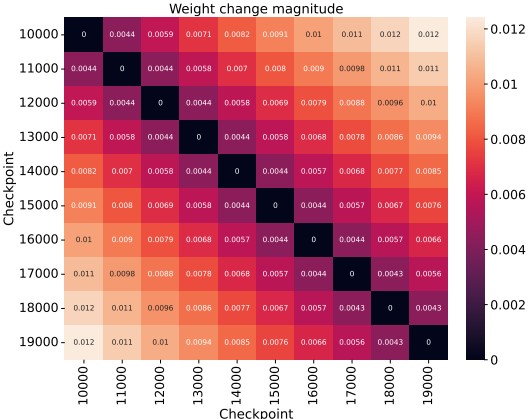

Figure 11: Distribution of weight changes of the model throughout training. Computed as the L2 distance between the flattened model weights at two different times during training

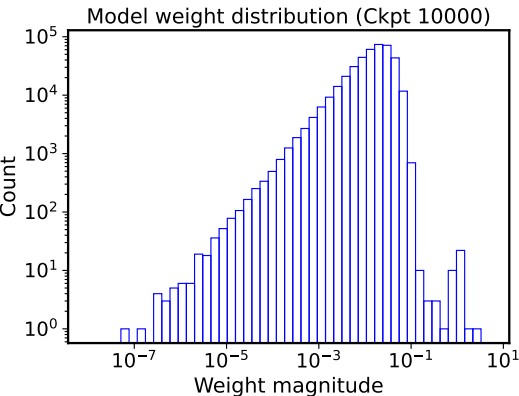

Figure 12: Distribution of weight magnitudes of the trained model at checkpoint 10k

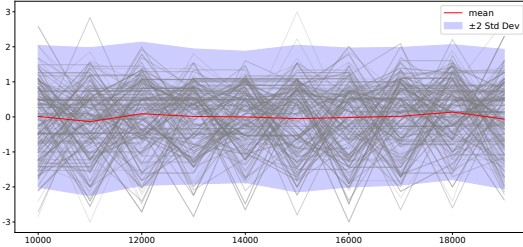

Figure 13: Dynamics of sequences with high complexity (>0.7) filtered by using maximum substring match of 30

