# OpenReview forum: "Uncovering Latent Memories in Large Language Models"
_ICLR.cc/2025/Conference — ICLR 2025 Poster_

### Official Review · Reviewer_TM5T · 2024-10-27

**Soundness:** 2
**Presentation:** 1
**Contribution:** 1
**Rating:** 3
**Confidence:** 4

**Summary:**

This paper discusses memorization in language models (LM) through training focusing on "latent" memorization when a model has _soft_ memorization strings during the training. The experiments measure memorization as a function of the dataset (complexity and repetitions, Figure 1), and training steps (using intermediate checkpoints, Figure 2). The paper also details a method to identify sequences that are "latent" memorized using a guess-and-check optimizer via adding random perturbations to weights and measuring memorization (Figure 4). Finally, the paper showcases that sequences with lower cross-entropy loss exhibit lower (latent) memorization.

**Strengths:**

- The paper discusses a fresh perspective on LM memorizing sequences with _latent_ memorization. This type of memorization feeds into the memorization at the checkpoint, which is finally considered to be deployed.

**Weaknesses:**

- The main claim of the paper that memorization is stationary throughout training (Section 3.2) is largely unsubstantiated in experimental results (Figure 3). There is no trend on the sample level to make an inference. It is unclear to me what the mean line entails here. Moreover, it is difficult to parse the experiment since axes are unlabelled and not clearly discussed in the Figure 3 caption.
- The variance of memorization comparison with that of random walk is not (directly) measured experimentally and analysis in Section 3.2 is unsupported.
- It is unclear how the claim that sequences are "memorized as often as they are forgotten" (line 255 in Section 3.2) is made.
- The assumption that the sequences with low duplication and high complexity (measured using zlib heuristic) are more likely to contain private/sensitive information makes sense, but experiments only cover latent memorization on highly complex sequences (zilb>0.8). Why not ablate on different complexity levels (like Figure 1) and see how results vary?
- The paper only focuses on syntactical memorization (Definition 2.1) with edit distance used as the only metric. It would be interesting to see the results for semantic memorization using soft embedding similar to methods like BERTScore [1].
- Adding random noise and checking if the edit distance is lower is a guess-and-check zeroth order optimization formulation. This is applied on the training/chosen dataset and thus this noise is not "random." This strongly opposes the hypothesis made that simply perturbing the weights with random noise produces similar effects as training.
- The latent memorization is poorly defined (also not explicitly) and oftentimes read as a circular definition. For example, samples recovered from an intermediate checkpoint are latent memorized (Section 3.3) but also the other way around.
- Limited experiments in terms of model sizes. All Pythia [2] models can be evaluated to see the trends across different sizes.
- Largely, throughout the paper, the results are unsubstantiated (as mentioned above) or anecdotal (Table 2 and Line 305). The writing is unclear and difficult to parse for the reader. The aim or main conclusion of the paper is thus unsupported and unclear.
---
[1] Tianyi Zhang, Varsha Kishore, Felix Wu, Kilian Q. Weinberger, & Yoav Artzi. (2020). BERTScore: Evaluating Text Generation with BERT.

[2] Stella Biderman, Hailey Schoelkopf, Quentin Anthony, Herbie Bradley, Kyle O’Brien, Eric Hallahan, Mohammad Aflah Khan, Shivanshu Purohit, USVSN Sai Prashanth, Edward Raff, et al. Pythia: A Suite for Analyzing Large Language Models Across Training and
Scaling. In International Conference on Learning Representations, 2023b

**Questions:**

1. Is the deduplication discussed in Section 3.2 applied to isolate the repetitions affecting memorization and focus only complexity of samples (as measured by zlib)? If so, is the same deduplication applied to training data as well (specifically for additional deduping discussed in line 244)?
1. What are the x-axis and y-axis in Figure 3a?
1. What does "the kl-LD of the sequences fluctuate (Figure 3) but do so in a way which is stationary across training" mean exactly in line 248?
1. Is latent memorization a general case of memorization (as measured by any chosen metric)?

---

> ### Author Response · Authors · 2024-11-16
> **Response to Reviewer TM5T (Part 1)**
>
> We would like to thank the reviewer for their thoughtful and critical feedback. We appreciate the reviewer’s acknowledgment of the novelty of our work. Next, responding to each concern in turn:
>
> >The main claim of the paper that memorization is stationary throughout training (Section 3.2) is largely unsubstantiated in experimental results (Figure 3). There is no trend on the sample level to make an inference. It is unclear to me what the mean line entails here. Moreover, it is difficult to parse the experiment since axes are unlabelled and not clearly discussed in the Figure 3 caption.
>
> We would like to clarify that our main claim is that there exists sequences in the training data which are memorized by the model, but are not visible under the current definitions of memorization, but are stored in the weights of the model. We show that these sequences are persistent, even though the model does not encounter them again. The stationarity of these sequences is evidence of this phenomena, and is demonstrated by observing that the memorization status of these sequences fluctuates as training progresses, but lacks a trend in the mean.
>
> As we described in the figure caption of 2a, “The red line denotes the mean … of the kl-LD of all sequences at a single point in time”. This is the population mean of all of the samples, and shows that at the population level, the memorization of these sequences is not decaying.
>
> We acknowledge that the y axis is missing in figure 3 and 2a. Figure 3 is a histogram so the y axis is the counts of the number of samples in each of the buckets. For 2a, we did include a brief description in the caption but axes will be added in future revisions.
>
> > The variance of memorization comparison with that of random walk is not (directly) measured experimentally and analysis in Section 3.2 is unsupported.
>
> We did not feel it was necessary to compare directly with a random walk because it is known that random walks have a variance which grows linearly with time. This is clearly not the case in our study. In order to further support this, we ran a variance ratio test and found that we were able to reject the null hypothesis that our data is generated from a random walk with a p value of <1e-8 for both Pythia-1b and Amber-7b models. We will include this test in future revisions.
>
> > It is unclear how the claim that sequences are "memorized as often as they are forgotten" (line 255 in Section 3.2) is made.
>
> In figure 3, we show the changes in memorization for different sequences. We showed that at the population level, approximately the same number of sequences became memorized as were forgotten, indicated by the symmetry of the distributions in each of the plots. This is further supported by figure 2a in which individual sequences are tracked through training and the lack of a general trend for these sequences.
>
> > The assumption that the sequences with low duplication and high complexity (measured using zlib heuristic) are more likely to contain private/sensitive information makes sense, but experiments only cover latent memorization on highly complex sequences (zilb>0.8). Why not ablate on different complexity levels (like Figure 1) and see how results vary?
>
> From our perspective, low-complexity sequences and repetitions act as confounders. There are structural aspects of low-complexity sequences which muddy the distinction between language/pattern learning and memorization which we sought to isolate in this study. For example, there are many training sequences that are simple repetitions of the same few tokens, or obvious continuations such as counting.  We did not include these simpler sequences because we felt that the model was able to reproduce these sequences without necessarily memorizing them.
>
> > The paper only focuses on syntactical memorization (Definition 2.1) with edit distance used as the only metric. It would be interesting to see the results for semantic memorization using soft embedding similar to methods like BERTScore [1]
>
> This could be an interesting direction, but our focus is on syntactical memorization because memorized PII is more harmful when memorized exactly - for example: phone numbers, addresses, social security numbers.

---

> ### Author Response · Authors · 2024-11-16
> **Response to Reviewer TM5T (Part 2)**
>
> > Adding random noise and checking if the edit distance is lower is a guess-and-check zeroth order optimization formulation. This is applied on the training/chosen dataset and thus this noise is not "random." This strongly opposes the hypothesis made that simply perturbing the weights with random noise produces similar effects as training.
>
> The purpose of this result is to demonstrate that sequences which are memorized are easily evoked from the model, which is why this zeroth order approach is compelling.
>
> The noise is random in the sense that no information about the training set is used to perturb the weights. It is true that we looked for specific sequences in the perturbed models which makes this approach unsuitable as an attack since the target sequences need to be known before-hand but our claim is that this random weight perturbation is not that dissimilar from the effects of progressive pre-training. We demonstrated this by comparing the distribution of kl-LD after additional pretraining with the distribution obtained from random perturbation (Figure 5a). This is a qualitative observation, but it supports the idea that these sequences are still encoded in the weights and can be revived with a small amount of effort.
>
> > The latent memorization is poorly defined (also not explicitly) and oftentimes read as a circular definition. For example, samples recovered from an intermediate checkpoint are latent memorized (Section 3.3) but also the other way around
>
> We defined latent memorization as sequences which appear not to be memorized at a certain point in time in training. (they cannot be detected using kl-style memorization techniques) but can be recovered either by subsequent training or random weight perturbation. We felt that this was explained in section 3.3 but we can add further detail if this is still confusing.
>
> > Limited experiments in terms of model sizes. All Pythia [2] models can be evaluated to see the trends across different sizes.
>
> We do plan on trying to understand how these results scale with the model size in future work, but we used models that were large since they were most similar to the current state of the art models used in language modeling.
>
> > Largely, throughout the paper, the results are unsubstantiated (as mentioned above) or anecdotal (Table 2 and Line 305). The writing is unclear and difficult to parse for the reader. The aim or main conclusion of the paper is thus unsupported and unclear.
>
> We include anecdotal evidence to aid the reader in seeing the details of how our metric works, as well as providing greater clarity into what types of changes happen in the model's output throughout training. These are not meant to be central to our claims in the paper, but rather provide more detail into the phenomenon that we observe.
>
> If you could help provide some examples where the writing is unclear, it would be appreciated in order to improve the paper.

---

> > ### Author Response · Authors · 2024-11-26
> >
> > We have updated our manuscript to include additional axes labels as well as include the variance ratio test in order to address some of the concerns raised. If you feel that your concerns have been addressed, we encourage you to update your score.

---

> > > ### Author Response · Authors · 2024-12-02
> > >
> > > Your review said:
> > > 1. The main claims of our paper were unsubstantiated, pointing to results showing that memorization is stationary and the distinction of memorization dynamics from a random walk
> > > 2. The choices to use high complexity sequences, focus on syntactical memorization, and apply random perturbations were incomplete or improper.
> > > 3.  You indicated that the writing is generally “unclear and difficult”
> > >
> > > The rebuttals/clarifications were:
> > >
> > > 1. That the stationarity of memorization is a secondary claim which is supported qualitatively and the distinction of the memorization status from a random walk is evidenced by the constant variance and our additional statistical testing showing the improbability of memorization status originating from a random walk process.
> > > 2. These experimental choices are appropriate for the setting and represents the most compelling setting in which this phenomena occurs
> > > 3. Requested additional clarity about where improvements can be made but have not received any specific feedback.
> > >
> > > We have not heard back from you during this discussion period and we ask that if you feel that these concerns have been addressed to update your score, or share more about what concerns remain.

---

### Official Review · Reviewer_psZH · 2024-11-04

**Soundness:** 3
**Presentation:** 4
**Contribution:** 3
**Rating:** 8
**Confidence:** 3

**Summary:**

This paper investigates memorization dynamics in large language models, focusing on "latent memorization" – a phenomenon where models retain complex sequences in memory even after minimal exposure. The authors demonstrate that these sequences, initially not retrieved verbatim, can later re-emerge in model outputs through weight perturbations, raising concerns about data privacy. The paper introduces a novel diagnostic test for identifying latent memorized sequences based on cross-entropy loss and suggests potential mitigations to enhance data privacy in model training. This work adds to ongoing conversations about data leakage and highlights the emerging challenges of memory persistence in AI models.

**Strengths:**

- **Originality**: The concept of latent memorization and the critique of kl-memorization provide new perspectives on memorization in LLMs, advancing our understanding of potential data privacy risks. TO further substantiate the novelty, authors could consider comparing with the recent findings related to verbatim memorization mechanisms, for example,  Huang et al.  (https://arxiv.org/pdf/2407.17817)
- **Quality**: The methodology is sound, with experiments designed to isolate memorization effects. The use of weight perturbations to reveal latent memories is innovative and strengthens the study's claims.
- **Clarity**: The writing and visualizations are clear, making complex ideas accessible and well-supported by empirical evidence.
- **Significance**: The findings are relevant to privacy and security in LLMs, addressing a gap in our understanding of how sequences can be memorized and later recalled. Huang et al.'s framework on verbatim memory could provide an anchor point for discussing how latent memorization behaviors extend verbatim patterns.

**Weaknesses:**

- **Scope of Evaluation**: The experiments are limited to specific language model architectures, raising questions about generalizability to other architectures and model sizes.
- **Mechanistic Explanation**: The authors suggest that latent memorization reflects stability in weight space, a hypothesis that could benefit from further clarification, particularly in light of Huang et al.’s findings that verbatim memorization relies on distributed states rather than specific model weights.

**Questions:**

1. Have the authors considered how latent memorization might vary with larger or smaller model sizes?
2. Could the proposed kl-Levenshtein distance metric be generalized for other memory-intensive tasks or model architectures?
3. Are there any specific limitations to the cross-entropy diagnostic when scaling up to larger models or applying it to real-world settings?

---

> ### Author Response · Authors · 2024-11-16
> **Response to Reviewer psZH**
>
> We would like to thank the reviewer for their thoughtful feedback and for highlighting the novelty of our work, particularly with regard to the concept of latent memorization and the critique of KL-memorization.
>
> For calibration purposes, we’d like to note that the ICLR 2025 rubric differs slightly from previous similar conferences. For example:
>
> - To indicate "Accept", the NeurIPS 2024 rubric says to use 7 whereas the ICLR 2025 rubric says to use 8
> - To indicate "Strong Accept", the NeurIPS 2024 rubric says to use 9 whereas the ICLR 2025 rubric says to use 10
>
> Next, responding to each concern in turn:
>
> > Have the authors considered how latent memorization might vary with larger or smaller model sizes?
>
> We have not considered how these properties scale in this work, but we are interesting in these questions in future work.
>
> > Could the proposed kl-Levenshtein distance metric be generalized for other memory-intensive tasks or model architectures?
>
> The kl-Levenshtein metric is agnostic to the model architecture and relies only on comparing the model’s output with a target sequence. Thus, it naturally is only applicable for text generation tasks.
>
> > Are there any specific limitations to the cross-entropy diagnostic when scaling up to larger models or applying it to real-world settings?
>
> Cross entropy is a widely used and efficiently computable metric that is not limiting computationally. It does require models for which cross entropy can be estimated which include almost all modern language models.

---

> ### Comment · Reviewer_psZH · 2024-11-27
>
> Thanks authors for their response. I don't have any further questions. I don't see any obvious problems in this paper and it's about an important topic, though I'm not an expert in this specific field. I'm raising my score from 6 to 8.

---

### Official Review · Reviewer_74QL · 2024-11-04

**Soundness:** 3
**Presentation:** 3
**Contribution:** 3
**Rating:** 6
**Confidence:** 3

**Summary:**

he paper investigates the phenomenon of memorization in large language models (LLMs), specifically focusing on complex sequences encountered only once during training. The authors highlight that certain sequences, termed "latent memories," can be memorized by the model and retained persistently throughout training, even without further exposure.

**Strengths:**

- The concept of "latent memorization" adds a significant new layer to understanding how LLMs retain and recall data.
- Providing a diagnostic for detecting latent memorization.

**Weaknesses:**

- The study primarily focuses on Pythia-1B and Amber-7B, limiting the generalizability to other, potentially larger, LLMs not tested.

**Questions:**

- Are there training modifications or strategies that could preemptively minimize or control latent memorization?
- How does the latent memorization phenomenon interact with fine-tuning? Would fine-tuning mitigate the presence of these latent memories?

---

> ### Author Response · Authors · 2024-11-16
> **Response to Reviewer 74QL**
>
> We would like to thank the reviewer for their thoughtful feedback and for highlighting the contribution of our work to understanding how LLMs recall information.
>
> For calibration purposes, we’d like to note that the ICLR 2025 rubric differs slightly from previous similar conferences. For example:
>
> - To indicate "Accept", the NeurIPS 2024 rubric says to use 7 whereas the ICLR 2025 rubric says to use 8
> - To indicate "Strong Accept", the NeurIPS 2024 rubric says to use 9 whereas the ICLR 2025 rubric says to use 10
>
> > Are there training modifications or strategies that could preemptively minimize or control latent memorization?
>
> We have not explored these questions in this work. Since our main takeaway from this paper is that the traces of training data may not be evident in the model responses, but in the weights of the model itself, it may be the case that even more sophisticated methods are able to extract more training data from the model weights. This draws a nice connection to the field of differential privacy which considers a more theoretical approach to this question.
>
> > How does the latent memorization phenomenon interact with fine-tuning? Would fine-tuning mitigate the presence of these latent memories?
>
> This is an interesting question which we have not explored here. It would boil down to whether the distribution shift inherent in fine-tuning is might be more disruptive than additional pre-training would be, although these effects might be mitigated by the method in which fine tuning is performed  such as LoRA training.

---

> > ### Comment · Reviewer_74QL · 2024-12-02
> >
> > Thank you for your feedback. I'll keep my original score.

---

### Official Review · Reviewer_qyKz · 2024-11-04

**Soundness:** 3
**Presentation:** 4
**Contribution:** 3
**Rating:** 8
**Confidence:** 4

**Summary:**

The paper studies the mechanics of memorization (as defined currently in LLM literature) in depth along two axes: number of repeats of a string during pretraining and complexity of string and largely focuses on high complexity strings that are exposed only once and thus, based on existing evidence, are less likely to be memorized. Based on this investigation the paper finds that (1) memorization of such strings is rather stable during training (ie there's no clear trend of catastrophic forgetting) and that (2) there is evidence of "latent" memorization, ie, sequences which the model has seen only once and seem to not be memorized, can be surfaced using simple modifications such as adding random noise to model weights. Since such latent memorization can go unnoticed by existing measures, the paper also shows that a simple evaluation of cross entropy loss on such sequences clearly shows lower loss for latent memorized sequences and could be a simple diagnostic to measure latent memorization.

**Strengths:**

-  The paper studies the very important problem of memorization, is well written and adds a fresh new perspective to this space.
 -  I appreciate that the authors use carefully constructed metrics to study this rather nuanced phenomenon. I appreciate the relaxation of Carlini et al's definition using Levenshtein distance since such a measure allows for careful analysis of what factors contribute and to what degree towards memorization.
 - It was also quite nice to see complexity of string being considered a a factor. There's always the question with compressible strings whether reconstructing them given context is memorization eg: given context "aabbaabb" if one were to logically extrapolate, "aa" would likely be the next two characters. At that point I'm not sure if calling it memorization is the right thing. So I appreciate that the authors focused on strings where such extrapolations are almost impossible.
 - The result with "latent" memorization is really neat and I really liked the mechanistic approach to arrive at the method of adding noise to weights to surface the latent memorized strings. This result also has many practical implications, eg, someone taking an off the shelf base model and then finetuning the model is running the risk of having a model that can regurgitate information that was exposed during pretraining -- something the entity finetuning has likely no control over.
 - The diagnostic for latent memorization is also very neat and intuitive, I wonder if cross entropy loss should be used for measuring all kinds of memorization, since ultimately "latent" memorization is also memorization.
 - Overall I really enjoyed reading this paper. I think it provides a fresh perspective on memorization and poses many interesting questions for the field.

**Weaknesses:**

See questions.

**Questions:**

- [Data statistics and memorization] I am a little confused by Section 3.1. You say "simpler strings are more easily memorized" -- but this could very well be an artifact of the checkpoints you use. Since you pick checkpoints not exactly at the beginning, I would suspect simpler kinds of in-context learning to be at play here. If strings are highly compressible (ie simpler on your z-complexity scale) I would imagine based on the "k" part of the prompt one can infer the "l" part without necessarily memorization. Secondly, your paper later on shows that this kl-LD definition of memorization can miss other kinds of memorization like latent memorization. So taking latent memorization into account, is the statement that "simpler strings are more easily memorized" still true?

 - [Stationarity of memorization] Maybe I missed this, but could you say how many tokens were seen in the window you consider for the stationarity analysis in eg Fig 2a? I wonder if stationarity is a function of how many tokens it sees in the considered window -- because surely a once-encountered string in training should have some limited shelf life? How many tokens does it take for the capacity in the model to run out to retain this latently memorized string?

 - [Mechanics of memorization] The intuition that the somewhat strange mechanics of memorization are related to weights is really intriguing to me. However, I wonder if the statement "but this is in conflict with the fact that the model weights are constantly evolving" is super convincing. Let's run with your hypothesis that strings are encoded in model weights, then to memorize a single 64 character string you'd need 32 parameters (assuming again the float16 setting in the paper and no redundancy in encoding the string in weights) out of more than a billion parameters overall. The chances that you are changing such a small fraction of weights significantly is, I think, rather small. Not absolutely necessary for this work but maybe future work should look at how weights change when a string goes from memorized to non-memorized (as per the kl-LD definition).

 - [Mechanics of memorization] since the memorized sequence is only encountered once, I would suspect when it's encountered also plays a role in these dynamics. You consider checkpoints that are still fairly early on in training (compared to more recent models like llama/mistral which train for much longer), what would happen if this encounter happened much later on when training has matured somewhat? If we are to run with your hypothesis that this memorization is encoded in the model weights, then I would suspect as you overtrain models, the capacity to keep such rare and hard strings latently memorized would decrease. Again perhaps something to look at in future work.

 - [Recovering latent memorized strings] I really liked the idea of weight perturbations, however I wonder if you could also pull another lever here, which is the "k" part of the context given to the model. That is, it would be very cool to know if perturbing the weights is the only way to get the model to surface up these latent memorized strings and whether some kind of prompt hacking couldn't get you the desired outcome? If indeed no amount of prompt hacking can get you to the "l" part of the string then I think this will be much stronger evidence for your hypothesis of these strings being encoded in the weights.

 - [General comment] Your proposed diagnostic of latent memorization based on CE loss could also be extended for "standard" memorization, right? Is there any reason you would see to prefer the kl-LD definition?

 - [General comment] An overall comment I have about the paper is that you *could* call this entire paper a critique of existing defintions of memorization. Ultimately "latent" memorization is also memorization and perhaps we should expand our current definition of memorization to also take into account things one could get the model to output given *any* modifications (eg the weight modifications you propose). Any reason where you think the kl-LD definition for memorization is still the "better" definition?

---

> ### Author Response · Authors · 2024-11-16
> **Response to Reviewer qyKz (Part 1)**
>
> We would like to thank the reviewer for their thorough and constructive feedback. We appreciate the acknowledgment of our use of Levenshtein distance and string complexity as nuanced approaches to investigating memorization. We’re also glad the reviewer found our mechanistic approach to latent memorization and the diagnostic method for uncovering memorized strings both useful and relevant to practical concerns.
>
> For calibration purposes, we’d like to note that the ICLR 2025 rubric differs slightly from previous similar conferences. For example:
>
> - To indicate "Accept", the NeurIPS 2024 rubric says to use 7 whereas the ICLR 2025 rubric says to use 8
> - To indicate "Strong Accept", the NeurIPS 2024 rubric says to use 9 whereas the ICLR 2025 rubric says to use 10
>
> Next, responding to each concern in turn:
>
> >[Data statistics and memorization] I am a little confused by Section 3.1. You say "simpler strings are more easily memorized" -- but this could very well be an artifact of the checkpoints you use. Since you pick checkpoints not exactly at the beginning, I would suspect simpler kinds of in-context learning to be at play here. If strings are highly compressible (ie simpler on your z-complexity scale) I would imagine based on the "k" part of the prompt one can infer the "l" part without necessarily memorization. Secondly, your paper later on shows that this kl-LD definition of memorization can miss other kinds of memorization like latent memorization. So taking latent memorization into account, is the statement that "simpler strings are more easily memorized" still true?
>
> This is definitely the case. Our use of the word memorization is the operational definition defined as the ability of the model to reproduce the target sequence verbatim. It is likely for simpler sequences, the model is able to utilize ICL to infer the remainder of the string. This is the motivation for using complex strings, in which these alternative strategies are not applicable, and it is clearer that the model must be doing memorization.
>
> > [Stationarity of memorization] Maybe I missed this, but could you say how many tokens were seen in the window you consider for the stationarity analysis in eg Fig 2a? I wonder if stationarity is a function of how many tokens it sees in the considered window -- because surely a once-encountered string in training should have some limited shelf life? How many tokens does it take for the capacity in the model to run out to retain this latently memorized string?
>
> The sequences we are observing fit well within the context window length, so this was not a concern for our model. Pythia-1B has a context window of 2048 tokens and Amber-7B also had a context window of 2048 tokens. The capacity of the model is more related to the number of weights, and the relationship between model capacity and weights is more complex due to the compressibility of language.
>
> > [Mechanics of memorization] The intuition that the somewhat strange mechanics of memorization are related to weights is really intriguing to me. However, I wonder if the statement "but this is in conflict with the fact that the model weights are constantly evolving" is super convincing. Let's run with your hypothesis that strings are encoded in model weights, then to memorize a single 64 character string you'd need 32 parameters (assuming again the float16 setting in the paper and no redundancy in encoding the string in weights) out of more than a billion parameters overall. The chances that you are changing such a small fraction of weights significantly is, I think, rather small. Not absolutely necessary for this work but maybe future work should look at how weights change when a string goes from memorized to non-memorized (as per the kl-LD definition).
>
> This is an interesting point. I think this is related to an interesting and challenging area of trying to localize how information is stored in the network and how it interacts with other training sequences. Generically, the gradient modifies all of the weights in the network which is classically why continual learning is difficult, but it seems that for these memorized sequences, they tend to be more persistent.

---

> ### Author Response · Authors · 2024-11-16
> **Response to Reviewer qyKz (Part 2)**
>
> > [Mechanics of memorization] since the memorized sequence is only encountered once, I would suspect when it's encountered also plays a role in these dynamics. You consider checkpoints that are still fairly early on in training (compared to more recent models like llama/mistral which train for much longer), what would happen if this encounter happened much later on when training has matured somewhat? If we are to run with your hypothesis that this memorization is encoded in the model weights, then I would suspect as you overtrain models, the capacity to keep such rare and hard strings latently memorized would decrease. Again perhaps something to look at in future work.
>
> We actually tried to pick a point in training where the model was trained for a while (~20% of the training run). It seemed like before this point in training, the model is still rapidly changing but at this point, memorization seems to more or less stabilize. It was noted in [1] that the position of the sequence in time doesn’t really affect whether it is memorized at the end of training, so we did not investigate how memorization changes after the checkpoint we used but it would be interesting to re-analyze this through the lens of latent memorization and also consider the effects of fine tuning and distribution shift.
>
> > [Recovering latent memorized strings] I really liked the idea of weight perturbations, however I wonder if you could also pull another lever here, which is the "k" part of the context given to the model. That is, it would be very cool to know if perturbing the weights is the only way to get the model to surface up these latent memorized strings and whether some kind of prompt hacking couldn't get you the desired outcome? If indeed no amount of prompt hacking can get you to the "l" part of the string then I think this will be much stronger evidence for your hypothesis of these strings being encoded in the weights.
>
> We did also try to train the model on each of the target sequences for a few gradient steps to see if latent memorized sequences are “re-learned” faster, but we found that cross entropy loss performed better at detecting these sequences which re-emerge than that method. This may be more related to the fact that the sequences which eventually re-emerge during training are more easily evokable, and “deeper” memorized sequences need more invasive methods to extract.
>
> In terms of prompt hacking, there has been some work along these lines which we included in our related work which are an interesting and alternative approach for extracting training data [2] [3]
>
> > [General comment] Your proposed diagnostic of latent memorization based on CE loss could also be extended for "standard" memorization, right? Is there any reason you would see to prefer the kl-LD definition?
>
> I think both are useful in practice, kl-LD matches our intuitive understanding of what memorization is better, so I feel that it is still a useful metric. It does seem like CE loss is a more robust measurement of memorization, but its less clear how this is connected to whether information can be extracted from the model and how that information might be used after it is extracted.
>
> > An overall comment I have about the paper is that you could call this entire paper a critique of existing definitions of memorization. Ultimately "latent" memorization is also memorization and perhaps we should expand our current definition of memorization to also take into account things one could get the model to output given any modifications (eg the weight modifications you propose). Any reason where you think the kl-LD definition for memorization is still the "better" definition?
>
> I think this is similar to the above question and I think the difficulty comes from the difficulty in defining memorization, as well as considering what risks arise from memorized training data.
>
> [1] Biderman, Stella, et al. "Emergent and predictable memorization in large language models." Advances in Neural Information Processing Systems 36 (2024).
>
> [2] Kassem, Aly M., et al. "Alpaca against Vicuna: Using LLMs to Uncover Memorization of LLMs." arXiv preprint arXiv:2403.04801 (2024).
>
> [3] Thakkar, Om, et al. "Understanding unintended memorization in federated learning." arXiv preprint arXiv:2006.07490 (2020).

---

### Meta-Review · Area_Chair_hg5e · 2024-12-20

**Metareview:**

The submission investigates the memorisation by large pre-trained models of sensitive information contained in training data sets. The main novelty of the paper is in analysing how the memorisation behaviour of LLMs is impacted by the complexity of sequences. Contrary to the current understanding, it is shown that complex string are in fact memorised throughout training (i.e., no catastrophic forgetting) and that adding random noise to model weights can reveal previously undetected memorised sequences. The reviewers mostly agree that this paper is novel, investigates a practically significant problem, and the investigation is undertaken in a sound and nuanced manner.

**Additional Comments On Reviewer Discussion:**

The authors did a good job of answering the large number of questions put forward in the original reviews. There was not much discussion from the reviewers, but there were some raised scores, indicating that the answers were informative.

---

### Decision · Program_Chairs · 2025-01-22

Accept (Poster)